# Combining Object-Based Image Analysis with Topographic Data for Landform Mapping: A Case Study in the Semi-Arid Chaco Ecosystem, Argentina

**Isabel Luisa Castillejo-González** [1,*] **, Cristina Angueira** [2] **, Alfonso García-Ferrer** [1] **and Manuel Sánchez de la Orden** [1]

1   Department of Graphic Engineering and Geomatics, Campus de Rabanales, University of Cordoba, 14071 Córdoba, Spain; agferrer@uco.es (A.G.-F.); ig1saorm@uco.es (M.S.d.l.O.)
2   National Institute of Agricultural Technology (INTA), Agricultural Experimental Station of Santiago del Estero, Santiago del Estero G4200, Argentina; cristina.angueira@gmail.com
*   Correspondence: ilcasti@uco.es; Tel.: +34-957-218-537

**Abstract:** This paper presents an object-based approach to mapping a set of landforms located in the fluvio-eolian plain of Rio Dulce and alluvial plain of Rio Salado (Dry Chaco, Argentina), with two Landsat 8 images collected in summer and winter combined with topographic data. The research was conducted in two stages. The first stage focused on basic-spectral landform classifications where both pixel- and object-based image analyses were tested with five classification algorithms: Mahalanobis Distance (MD), Spectral Angle Mapper (SAM), Maximum Likelihood (ML), Support Vector Machine (SVM) and Decision Tree (DT). The results obtained indicate that object-based analyses clearly outperform pixel-based classifications, with an increase in accuracy of up to 35%. The second stage focused on advanced object-based derived variables with topographic ancillary data classifications. The combinations of variables were tested in order to obtain the most accurate map of landforms based on the most successful classifiers identified in the previous stage (ML, SVM and DT). The results indicate that DT is the most accurate classifier, exhibiting the highest overall accuracies with values greater than 72% in both the winter and summer images. Future work could combine both, the most appropriate methodologies and combinations of variables obtained in this study, with physico-chemical variables sampled to improve the classification of landforms and even of types of soil.

**Keywords:** data mining algorithms; DEM-derived variables; geoforms classification; Landsat-8 imagery; OBIA

## 1. Introduction

The Gran Chaco or Dry Chaco is a flat, semi-arid ecosystem characterized by a mix of woodlands and grasslands. The Argentine Chaco is undergoing detrimental transformations due to unrestricted forest clearing and fire because of activities ranging from traditional land use to commercial agriculture [1]. To mitigate the impacts caused by agricultural expansion, a balanced environmental performance is required, which involves obtaining accurate information of the factors that affect the land [2]. Nevertheless, the reduced number of soil studies and the limited trained staff show the lack of adequate soil data to aid informed planning of land use, which is threatening the sustainable development of the region.

Conventional soil mapping techniques are quite expensive and time consuming. Soil types are usually delineated in the field, based on the relationship of soils and their natural surroundings,

following tacit mental models [3]. Those models are rarely described with clarity [4], are subject to personal bias, and are difficult to replicate, especially in quantitative studies [5]. However, the increasing availability of data combined with the rapid development of new information processing tools is significantly changing the way in which soil and other geoforms like landscapes, moldings or landforms are being mapped. The digital soil mapping (DSM) techniques combine field observations, laboratory measurements, terrain variables and remote sensing data, integrated with quantitative methods to map spatial patterns of soil properties [6]. This combination of techniques facilitates mapping inaccessible areas or areas with economical restrictions by reducing the need for extensive field surveys. As landforms can be suitable predictors of soil types as soil development often occurs in response to the underlying lithology and water movement in the landscape [7,8], the application of these techniques for the classification of landforms can be consider as the first stage of a future soil mapping procedure.

Most of the studies based on DSM techniques combine multispectral satellite data with topographic data to improve geoform classifications, especially in complex landscapes [9–11], whereas traditional image analysis techniques use pixel-based classification approaches. However, when medium spatial resolution imagery is used in large areas, especially in land characterized by high intra-class spectral variability, analyzing pixels individually can produce misclassifications. One possible solution is to apply Object-Based Image Analysis (OBIA) to group adjacent pixels into spectrally and spatially homogeneous objects created through a segmentation process. Although there are different types of OBIA analyses, the use of the multiresolution segmentation algorithm has been found the most sensitive to morphological discontinuities in DEMs [12] showing a great ability of capturing morphological discontinuities in natural spatial entities such as landforms [13]. OBIA techniques have shown a great potential for classifying compared to pixel-based methods in agriculture [14,15], forestry [16,17] and urban areas [18,19], among other disciplines. For landform classification, the analysis of satellite and topographic data with object-based approaches offered promising results in steep [20–22] and deltaic areas [23]. Nevertheless, as far as we know, the use of this technique in large areas of semi-arid ecosystem have not been evaluated.

Considering that, the main goal of this project was to establish a multidisciplinary OBIA approach to discriminate and map landforms in the semi-arid Chaco ecosystem in order to automate the process with spectral and topographic data. For this purpose, different conceptual and mathematical classification algorithms were tested.

## 2. Materials and Methods

### 2.1. Study Area

The study area is approximately 8800 km$^2$ (110 km × 80 km) located in the center of the Santiago del Estero province, Argentina (between 27°30′ S–28°35′ S and 63°45 W–64°35′ W, datum WGS84) (Figure 1). This area is representative of the Chaco ecosystem, characterized by a continental subtropical climate with dry and mild winters and summers marked by extreme high temperatures. Precipitation is concentrated during summer. Comprising part of the fluvio-eolic Chaco Plain, Río Dulce and Río Salado alluvial Plains, the territory slopes gently from west to east and exhibits a set of landforms resulting from exogenous and endogenous processes.

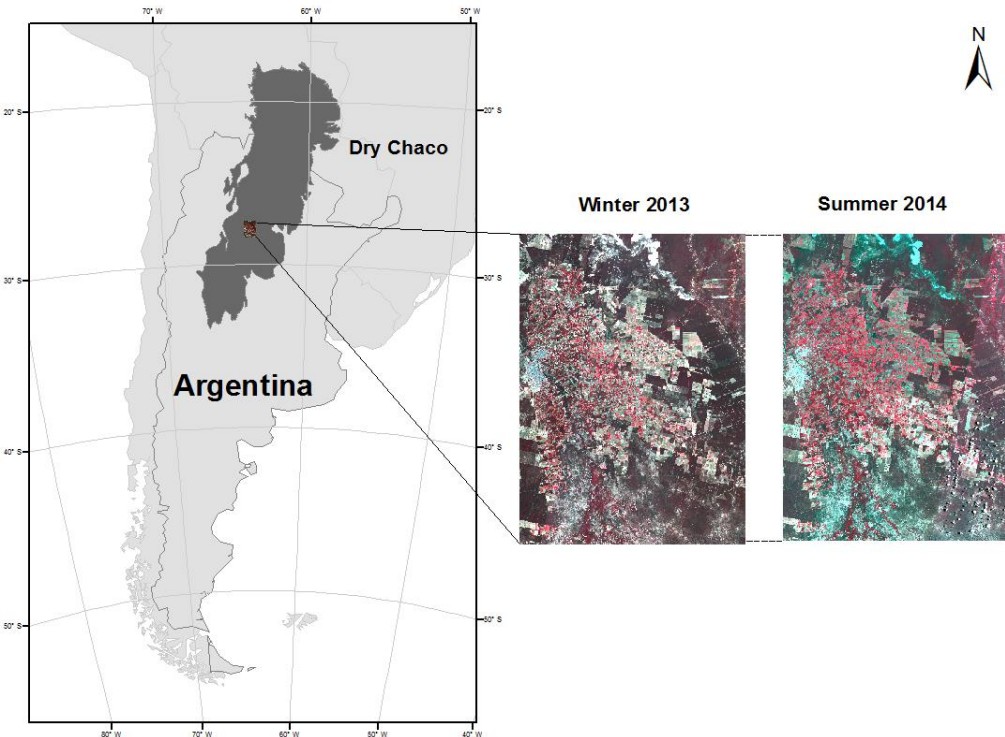

**Figure 1.** Location of the study area in the semiarid Chaco, province of Santiago del Estero, Argentina. The study area is depicted by the winter 2013 and summer 2014 Landsat multispectral image.

## 2.2. Satellite Data and Preprocessing

In the winter of 2013 and the summer of 2014, digital imagery data of the study region were acquired by the Landsat 8 satellite (winter-13 and summer-14 imagery, respectively). Seven of the eight multispectral Landsat 8 OLI bands (B1: 435–451 nm, B2: 452–512 nm, G: 533–590 nm, R: 636–673 nm, NIR: 851–879 nm, SWIR1: 1566–1651 nm, SWIR2: 2107–2294 nm) were used in this study. The spatial resolution was 30 meters and the radiometric resolution was 16 bits. The bands analyzed presented a preprocessing level L1T, with standard terrain corrections that provides radiometric and geometric accuracy through the use of digital elevation models and ground control points. Both images were atmospherically corrected using the Fast Line-of-sight Atmospheric Analysis of Hypercubes (FLAASH) method, where the images were first calibrated to at-sensor radiance, and then corrected to surface reflectance. The study area includes the city of Santiago del Estero and surrounding agricultural areas, where the original land uses have been altered by human intervention. As spectral signatures of altered surfaces do not contain relevant information for soil and terrain mapping, this area was masked in the study as Mulder et al., [24] recommends.

## 2.3. Ground-Truth Landform Distribution

To substantiate and validate the classification procedures, ground-truth landforms samples were defined from the geopedologic map at a scale of 1:500,000 carried out by Angueira [25]. This map was developed based on recognition-intensive sampling where physical and chemical data obtained from soil profiles were analyzed and related with topographic and other ancillary information. This process was developed based on the knowledge of the technical staff and took several years to complete.

In the study area, different features can be distinguished: a fan with its apex in the west and a divergent gently sloping area to north-east and south-east, a main and secondary fault, two north-south parallel valleys at the foot of the main fault, and a shallow sag pond at the foot of the secondary fault. As a result, the study area comprises of three landscapes types, which were divided into nine moldings and fourteen homogeneous and mutually contrasting landform units (Table 1).

The Fluvio-Eolian Chaco Plain is a slightly convex landscape with 0.5–1% slope of the Sali-Dulce River system. This landform includes three moldings: (a) the proximal megafan, a gently sloping terrain covered by a loess mantle including an gently sloping loess cover unit (1P) dissected by blowout depressions (2P) which are wide and elongated shallow areas occasionally acting as runoff paths; (b) the distal megafan, a cone-shaped deposit of sand and finer materials formed in the area where the river slows down and spreads into a flatter plain at the exit of the Huymampa N-S fault, including a flat or gently sloped interfluvial plain (3P) and an irregular flat with elongated or curvilinear shallow depressions infilled channel (4P) backfilled with sediment and located within the interfluvial plain; (c) the old alluvial overland flow area, with a relatively flat overflowed depression (5P), a nearly closed fan-shaped accumulation of deposit of sand-sized and finer sediments formed where the currents slow and dissipate, crossed by many channel and levees, elongated parallel areas oriented NW-SE crossing the slightly lower flood plain, named Alluvial overfluvial levee (6P).

The Valley (Dulce River) landscape is characterized with terraces and watercourses and includes two moldings: (a) the middle terraces, the higher ground area formed by the river on its right side with levee and former watercourse (7D), (b) a low terrace identified at the left side of the river, showing flat surfaces that border lie above the flood plain formed from the deposition of alluvium adjacent to the river that periodically overflows (8D); and (c) an active floodplain landscape formed by the main course of the Dulce River and lower order courses generally dry, leading floodwaters spilling out of the riverbed (9D).

The Alluvial migratory plain (Salado River) is an flat area composed by three moldings: (a) an active fluvial valley formed by an extensive, depressed area between natural levees and terraces (10S) and elongated high areas, almost parallels in north-south direction distributed throughout the alluvial overflow plain (11S); (b) an active floodplain including streams and a low, saturated ground, intermittently covered with water and vegetated by shrubs and trees that drains excess system marshes and lagoons (12S); and (c) a fluvioeolic terrace remnant formed by a large gently sloping area, nearly level, erosional remnant of an alluvial plain without drainage network (13S), and a concave shallow microrelief through which runoff is drained in periods of high water (14S).

Landform, a basic geoform type characterized by a unique combination of geometry, dynamics and history, was chosen to be classified due to its close relationship with soils. Of the fourteen landforms, only thirteen were considered in the classification process due to the 4P landform was insignificant in the unmasked study area.

**Table 1.** Ground-truth information obtained from a geopedologic survey [19].

| Landscape | Molding | Facies | Landform | Code |
|---|---|---|---|---|
| Fluvio-eolian Chaco plain (Sali-Dulce River) | Proximal megafan | Eolian | Loess cover<br>Blowout depression | 1P<br>2P |
| | Distal megafan | Alluvial | Interfluvial plain<br>Infilled channel | 3P<br>4P |
| | Old alluvial overland flow | Alluvial | Overflowed depression<br>Alluvial overflow levee | 5P<br>6P |
| Valley (Dulce River) | Middle terrace (mt) | Alluvial | Levee and overflows (mt) | 7D |
| | Low terrace (lt) | | Levee and overflows (lt) | 8D |
| | Active floodplain | | River | 9D |
| Alluvial migratory plain (Salado River) | Active fluvial valley | Alluvial | Alluvial overflow plain<br>Levee | 10S<br>11S |
| | Active floodplain | Alluvial | Alluvial overflow swamp | 12S |
| | Fluvio-eolian terrace remnant | Eolian over alluvial | Alluvial flat<br>Alluvial channel | 13S<br>14S |

Two systematic samples with approximately 4858 ha distributed over the different soils were performed over the entire surface to collect ground-truth information (Figure 2a). A total of 4969 points separated by 930 m each were used to collect the spectral signature in the algorithmic training process

(Figure 2b). The remaining 49,011 points separated by 300 m were used to assess the accuracy of the classifications (Figure 2c). This procedure sampled pixels within each class proportionally to the extent of the soil units.

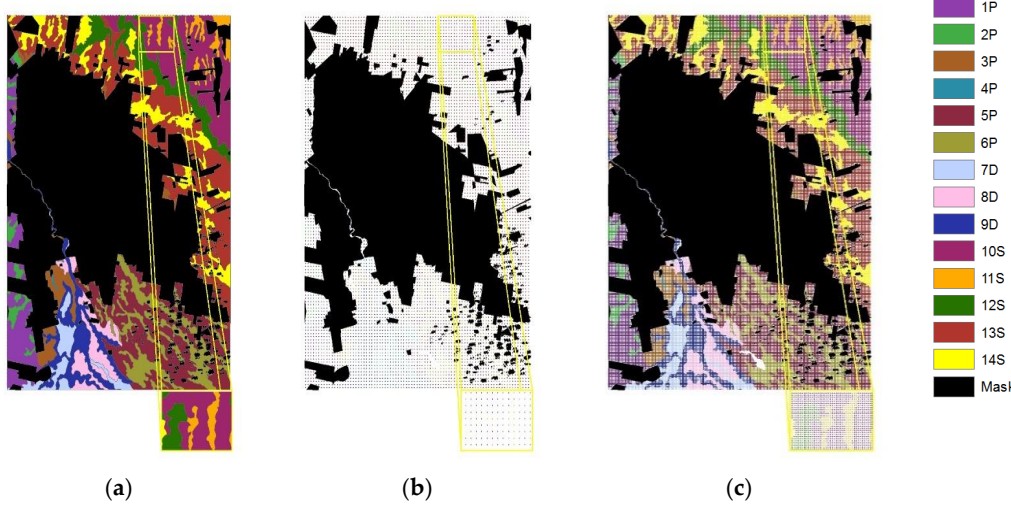

(**a**)　　　　　　　　　　　　　　　(**b**)　　　　　　　　　　　　　　　(**c**)

**Figure 2.** Distribution of georeferenced samples: (**a**) ground-truth information; (**b**) training samples and (**c**) validation samples.

## 3. Methods

To evaluate the accuracy of the delimitation among the different landform units, two stages of classification were performed. The methodology flowchart is shown in Figure 3.

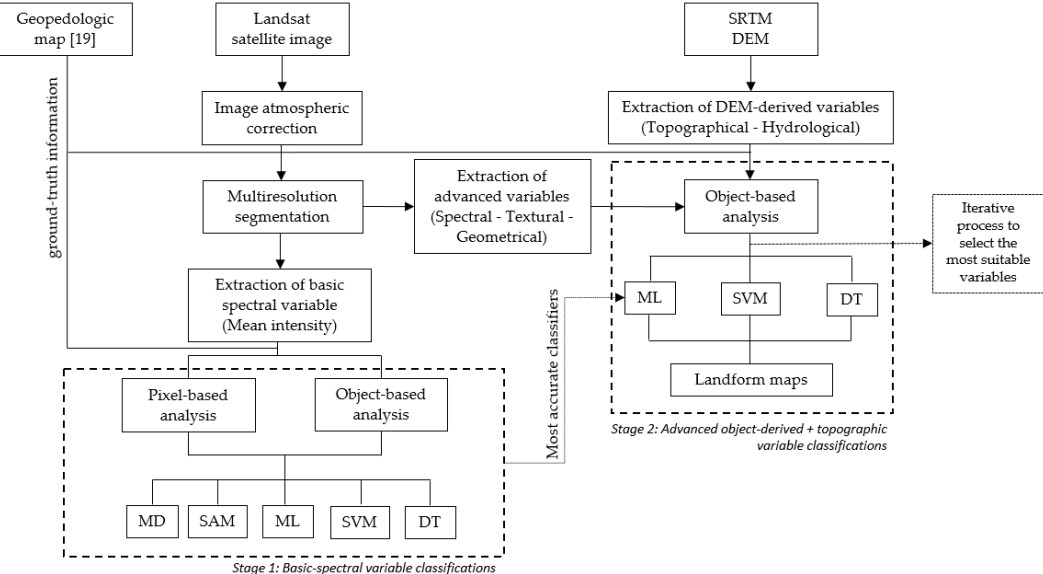

**Figure 3.** Processing scheme for the classification of landforms combining object-based image analysis with topographic data.

### 3.1. Stage 1: Basic-Spectral Variable Classifications

The goal of the basic-level study was to compare the accuracy of pixel- and object-based classifications of Landsat 8 bands using different classifiers in order to determine whether the segmentation of the image would improve the accuracy of landform maps.

### 3.1.1. Segmentation

The seven multispectral bands were partitioned into homogeneous objects using the Fractal Net Evolution Approach (FNEA) segmentation algorithm. This algorithm allows for multiresolution segmentation which groups pixels and creates highly homogeneous image objects while minimizing the average heterogeneity at an arbitrary resolution. The objects created can be used as the base of classification and other processing procedures [26].

During the segmentation process, the weighting of input data and parameters such as scale, color and shape must be controlled. The scale parameter determines the maximum allowed heterogeneity for the resulting image objects. The color and the shape (smoothness and compactness) parameters define the percentage that the spectral values and the shape of objects, respectively, will contribute to the homogeneity criterion. To obtain the segmentation parameters with the greatest precision for each study, their ability to accurately delineate landforms units in various scenarios must be evaluated. In this study, several parameters were tested. The most satisfactory combination for scale, color, shape, smoothness and compactness values in the winter-13 image were 650, 0.9, 0.1, 0.5 and 0.5 and in the summer-14 image were 450, 0.9, 0.1, 0.5 and 0.5, respectively (Figure 4).

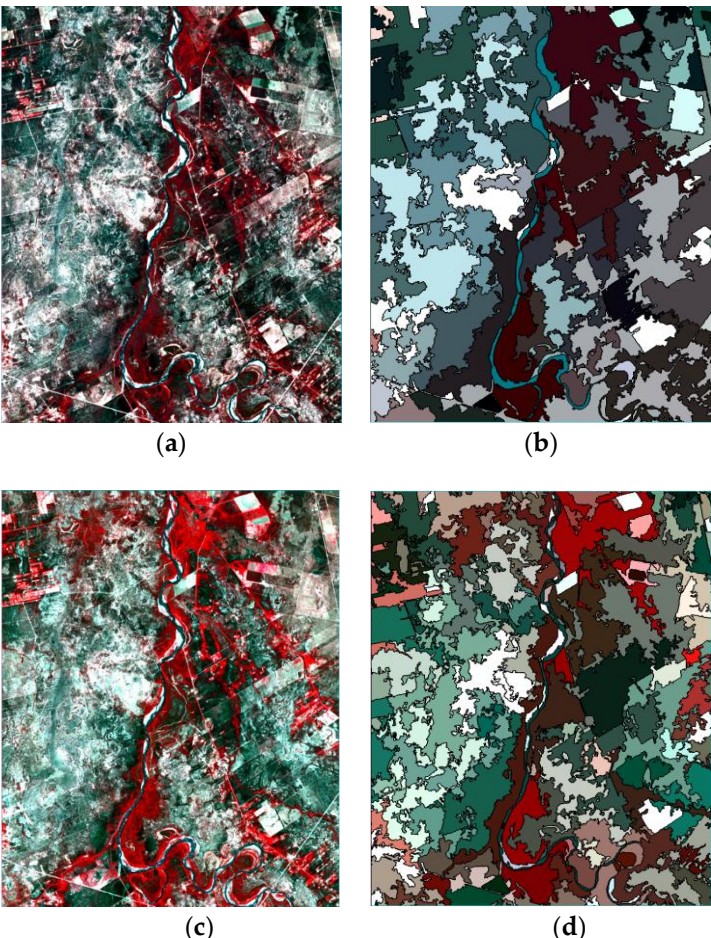

**Figure 4.** Detail of Landsat 8 imagery: (**a**) winter 2013 and (**c**) summer 2014 pixel imagery. Segmentation into separate objects in (**b**) winter 2013 and (**d**) summer 2014.

### 3.1.2. Basic-Spectral Landform Classification and Accuracy Assessments

For both pixel- and object-based analysis, different supervised classification algorithms were used to classify the Landsat 8 images collected in 2013 and 2014. Five classification methods were chosen due to their strong conceptual and mathematical differences: Mahalanobis Distance (MD), Spectral

Angle Mapper (SAM), Maximum Likelihood (ML), Support Vector Machine (SVM) and Decision Tree (DT). The MD and SAM classifiers assign a class to a pixel when the minimum spectral distance or the minimum spectral angle from that pixel to the spectral distance or spectral angle average of the distinct classes are found, respectively. ML is a probability classifier that selects the set of values from the model parameters that maximizes the likelihood function and allocates each pixel to the class with which it has the highest posterior probability of membership. Finally, the SVM and the DT methods analyze data and recognize patterns via data mining. For the SVM classification, the radial basis function (RBF) kernel was chosen, as recommended by Hsu et al. [27]. The RBF kernel is a Gaussian non-linear function, which constructs a set of hyperplanes in a high- or infinite-dimensional work space. The DT classification performed in the study was carried out by the C4.5 algorithm [28], which is a top-down inductor of decision trees that expands nodes in depth-first order for each step using the divide-and-conquer strategy. The accuracy of the methods was analyzed by quantifying coincidences between the estimated map and ground-truth with a confusion matrix analysis and a Kappa test. The confusion matrix provides the overall accuracy (OA) of the classification, which indicates the percentage of correctly classified pixels. On the other hand, the Kappa test (K) determines whether the results presented in the confusion matrix are significantly better than random or chance classification, indicating a more conservative estimation than simple percent agreement value. The Equations used to calculate the OA (1) and K (2) are given as follows:

$$OA = \frac{\sum_{i=1}^{r} x_{ii}}{n} \tag{1}$$

$$K = \frac{n \sum_{i=1}^{r} x_{ii} - \sum_{i=1}^{r} x_{i+} x_{+i}}{n^2 - \sum_{i=1}^{r} x_{i+} x_{+i}} \tag{2}$$

where r is the number of rows in the matrix, $x_{ii}$ is the number of observations in row *i* and column *i*, $x_{i+}$ and $x_{+i}$ are the marginal totals of row *i* and column *i*, respectively, and n is the total number of observations.

In order to be compared without any subjective estimation, the data classified in pixel- and object-based analyses must be similar. While the seven original multispectral bands were analyzed in the pixel-based classifications, in object-based analyses the bands classified were the mean of the intensity values of the objects obtained in the segmentation of the multispectral bands.

### 3.2. Stage 2: Advanced Object-Derived + Topographic Variable Classifications

In this analysis, the main goal was to obtain accurate maps of landforms with OBIA approaches using the most accurate classification algorithms identified in the previous stage. Different object-based classifications were carried out to test combinations of advanced spectral, textural and geometrical object-derived variables. Topographic ancillary data were also included in this level to provide useful information to improve the classification process.

### 3.2.1. Object-Derived Variables

Segmentation can generate a large quantity of object metrics for a given image. In this study three types of object metrics were used: spectral, textural and geometrical (Table 2). Five spectral features were evaluated to describe image objects with the information derived from their spectral properties. The mean feature, being the first variable tested in all classifications, was used as the main parameter for the classification. Additionally, 4 textural and 14 geometrical more features obtained from the segmented objects were added to the spectral information. The textural parameters, calculated based on Haralick's co-occurrence matrix, were used to detect the local differences between objects by analyzing how often different combinations of pixel gray levels occur throughout the images. The geometrical information measures the characteristics of the shape of the segmented object and was calculated from the pixels that form each object.

**Table 2.** Advanced object-based variables derived from segmentation of Landsat 8 images.

| Type | Variable | Name | Brief Description |
|---|---|---|---|
| Spectrality | Mean | Mean | Mean of the intensity values of all pixels forming an image object |
| | Standard deviation | St_Dev | Standard Deviation of the intensity values of all pixels forming an image object |
| | Skewness | Skew | Asymmetry of the distribution of all pixels forming an image object |
| | Brightness | Bright | Sum of brightness weight in all layers of an image object multiply by the mean intensity of the same object |
| | Max difference | Max_Diff | Ration between the maximum difference of mean intensity of an image object in the different layers and the brightness of the same image object |
| Texture | Correlation | GLCM_C | Linear dependency of gray levels of neighboring pixels on the gray level co-occurrence matrix (GLCM) |
| | Entropy | GLCM_E | Distribution of the pixel values on the gray level co-occurrence matrix (GLCM) |
| | Homogeneity | GLCM_H | Amount of local variation in the image based on the gray level co-occurrence matrix (GLCM) |
| | Mean | GLCM_M | Average expressed by the frequency of occurrence of a pixel combination with a certain neighbor pixel value |
| Geometry | Area | Area | Number of pixels forming an image object |
| | Length | Length | Multiplication between the number of pixels and the length-to-width ratio of an image object |
| | Width | Width | Ration between the number of pixels and the length-to-width ratio of an image object |
| | Asymmetry | Asymm | Relative length of an image object compared to a regular ellipse polygon |
| | Border index | Border_I | Ratio between the border lengths of the image object and the smallest enclosing rectangle |
| | Compactness | Compact | Ratio between the length x width of the object and its area |
| | Density | Density | Ratio between the area of an image object and its approximated radius |
| | Elliptic fit | Ellip_Fit | Comparison between the area of an imagen and an ellipse with the same area as the selected image object |
| | Main direction | Main_Dir | Direction of the eigenvector belonging to the larger of the two eigenvalues |
| | Radius of largest enclosed ellipse | R_Largest | Ratio of the radius of the largest enclosed ellipse to the radius of the original ellipse |
| | Radius of smallest enclosing ellipse | R_Smallest | Ratio of the radius of the smallest enclosing ellipse to the radius of the original ellipse |
| | Rectangular fit | Rect_Fit | Comparison between the area of the image object outside a rectangle with the same area as the image object, and the area inside the rectangle |
| | Roundness | Round | Difference of the enclosing ellipse and the enclosed ellipse |
| | Shape index | Shape_I | Comparison between the border length feature of the image object and four times the square root of its area |

### 3.2.2. Topographic variables

The basic morphometric identifying parameters are derived from digital elevation model (DEM) [24]. The Shuttle Radar Topographic Mission (SRTM) terrain data, a 30 m elevation model, provided the elevation values for the study and the basis for the calculation of the derived topographic variables. The 11 DEM-derived variables were divided in two groups (Table 3). Five DEM-derived attributes were direct descriptors of the landforms representing the morphological attributes of the terrain. This group included the terrain elevation and the first and second derivative. The second group of DEM-derived variables characterized specific hydrological processes that were identified to be important soil-forming discriminators. This group included terrain indices that combine two or more terrain attributes (compound derivatives).

**Table 3.** DEM-derived variables from The Shuttle Radar Topographic Mission (SRTM) terrain data.

| Type | Variable | Name | Brief Description |
|------|----------|------|-------------------|
| Topography | Elevation | Elev | Terrain altitude on a reference system |
| | Slope | Slope | Steepness of the terrain relative to the horizontal plane |
| | Aspect | Aspect | Compass the direction that a terrain slope faces |
| | Plan Curvature | Plan_Cuv | Curvature of the hypothetical contour line that passes through a specific cell |
| | Profile Curvature | Prof_Curv | Curvature of the surface in the direction of the steepest slope |
| Hydrology | Altitude about channel network | Alt_Ch | Vertical distance to a channel network base level |
| | Catchment area | Catch_Area | Area of land draining into a stream or a water course |
| | Channel network base level | Ch_Net | Base level of a channel network |
| | Convergence index | Conv_I | Structure of the relief as a set of convergent areas (channels) and divergent areas (ridges) |
| | LS Factor | LS_Factor | Combination of slope length factor (L) and slope steepness factor (S) to compute the effect of slope length and slope steepness on erosion. |
| | Wetness index | Wet_I | Value in a flow accumulation raster for the corresponding DEM |

### 3.2.3. Advanced Landform Classification and Accuracy Assessments

The object-based and topographic variables were classified using those classification methods that yielded more accurate results in the previous stage of study: the ML, SVM and DT classification algorithms. In order to obtain an accurate combination of variables to distinguish landforms, the variables were added to the procedure in an ordered manner. In each test, if the new variable did not offer enough information to the study, that variable was eliminated from the analysis. First, only spectral variables were tested. Then, the topographic variables were combined jointly with the most efficient spectral combination (S) to obtain a more accurate combination of spectral + topographic variables (S + To). That combination was evaluated with the textural variables. The most accurate combination of spectral + topographical + textural variables (S + To + Tx) was then analyzed with geometrical variables, resulting in the spectral + topographical + textural + geometrical combination (S + To + Tx + G) with the greatest precision. Additionally, alongside the OA and the K values used to show the accuracy of the classifications, the producer's accuracy (PA), which indicates the probability that a classified pixel actually represents that category in reality, was also evaluated. The Equation used to calculate the PA (3) is given as follows:

$$PA = \frac{x_{ii}}{x_{i+}} \tag{3}$$

where $x_{ii}$ is the number of observations in row $i$ and column $i$ and $x_{i+}$ is the marginal total of row $i$.

### 3.2.4. Evaluation of the Importance of the Variables in the Prediction

The 34 variables used to predict landform units showed different levels of contribution in obtaining the most accurate classification for each classifier and for both images. While some of

these variables were not ultimately used for the classification, it was necessary to test each variable comparatively. A first analysis was carried out to detect which variables were to be used in obtaining the most accurate classifications and the weight of each group of variables in each classification. The level of utility was quantified by analyzing the number of times each variable was used in the most accurate classifications obtained with the three different algorithms applied in two different images. A second analysis was performed only with the most accurate DT classification of each image. Whereas the ML and SVM algorithms give equal weight to all the variables involved in the classification process, each variable in the DT algorithm intervenes with a different utility in the prediction model. This contribution was assessed through the quantification of the number of nodes of the tree where each variable is used.

For Stage 1 and Stage 2 studies, the software eCognition Developer 8 (Definiens AG, 2009) was used to obtain the segmented bands and the object-based derived variables. ArcGIS 10.5 (ESRI, 2017) and SAGA GIS (SAGA GIS, 2017) extracted all the DEM-derivate variables (topographical variables). With Weka 3.8 software (University of Waikato, Hamilton, New Zealand, 2017), the decision tree sequences were obtained. The ENVI 5.1 software (ITT Virtual Information Solutions, 2013) was used to perform the atmospheric FLAASH correction, all the pixel- and object-based classifications and accuracy analyses.

## 4. Results

### 4.1. Stage 1: Basic-Spectral Variable Classifications

Accuracy assessments of the discrimination of the thirteen landforms in pixel- and object-based classifications are shown in Table 4. Of the 20 classifications evaluated, all the object-based analyses yielded better results than the pixel-based analyses. Table 4 reveals consistent differences among the classification algorithms. All pixel-based classifications showed low accuracies with OA values ranging from 23% to 42.6% in the winter-13 image, and from 19.4% to 36.8% in the summer-14 image. The least accurate classification method was SAM and most accurate was SVM. In the object-based analyses accuracy increased significantly, over 12%, for most of the classifiers evaluated. While the SAM algorithm continued having the lowest accuracies with OA values at 28.8% and 22.5% in the winter-13 and summer-14 images, respectively, the DT classifier surpassed the SVM algorithm, obtaining the highest accuracies of all the object-based classifications. DT yielded values of OA at 67.2% in winter-13 image and 71.5% in summer-14 image. Nevertheless, the highest OA value of the SVM object-based analysis was 55.1% for the winter-13 image. MD and ML showed moderate accuracies in both analyses, although ML generated more precise object-based classifications than MD. The highest performance values for OA and K for most of the classifiers occurred in the winter-13 image except for the DT object-based classification, which achieved the maximum OA and K with the summer-14 image.

**Table 4.** Classification accuracies of landforms in winter and summer images using different classification algorithms.

| | MD [1] | | SAM | | ML | | SVM | | DT | |
|---|---|---|---|---|---|---|---|---|---|---|
| | OA [2] | K | OA | K | OA | K | OA | K | OA | K |
| **Winter-13 Image** | | | | | | | | | | |
| **PBIA [3]** | 31.7 | 0.25 | 23.0 | 0.16 | 32.1 | 0.25 | 42.6 | 0.33 | 38.3 | 0.30 |
| **OBIA** | 43.7 | 0.37 | 28.8 | 0.22 | 49.2 | 0.43 | 55.1 | 0.48 | 67.2 | 0.63 |
| **Summer-14 Image** | | | | | | | | | | |
| **PBIA** | 28.7 | 0.22 | 19.4 | 0.13 | 29.5 | 0.23 | 36.8 | 0.25 | 36.5 | 0.28 |
| **OBIA** | 40.7 | 0.34 | 22.5 | 0.16 | 46.1 | 0.40 | 46.2 | 0.37 | 71.5 | 0.68 |

[1] Methods of classification. MD: Minimum Distance; SAM: Spectral Angle Mapper; ML: Maximum Likelihood; SVM: Support Vector Machine; DT: Decision Tree (J48 algorithm). [2] Accuracy values. OA: overall accuracy (%), K: Kappa coefficient. Type of analysis. PBIA: Pixel-based image analysis; OBIA: Object-based image analysis.

*4.2. Stage 2: Advanced Object-Derived + Topographic Variable Classifications*

Table 5 summarizes the most accurate classification results, OA and K, obtained from the three classification algorithms tested with spectral, topographical, textural and geometrical variables. The classifiers analyzed yielded medium classification accuracies, although slightly lower OA and K values were obtained in the winter-13 image than in summer-14 image. Only the SVM algorithm showed better results with the S and S + To combination of variables in the winter-13 analyses. The table reveals consistent differences, around 13.5%, in OA when comparing the most and the least accurate classifier with the most efficient combination of variables in both images. Thus, in the winter-13 image the greatest OA was 58.7% for ML (Figure 5a), 64.5% for SVM (Figure 5b) and 72.0% for DT (Figure 5c). Greater OA values were obtained for the summer-14 image with values of 60.8%, 64.8% and 74.7% for MP (Figure 5a'), SVM (Figure 5b') and DT (Figure 5c'), respectively.

**Table 5.** Best classification accuracies obtained with combination of advanced object-based and topographical variables in winter and summer images using Maximum Likelihood, Support Vector Machine and Decision Tree algorithms.

| Variables | Winter-13 Image | | | | | | Summer-14 Image | | | | | |
| --- | --- | --- | --- | --- | --- | --- | --- | --- | --- | --- | --- | --- |
| | ML [1] | | SVM | | DT | | ML | | SVM | | DT | |
| | OA [2] | K | OA | K | OA | K | OA | K | OA | K | OA | K |
| S [3] | 52.9 | 0.47 | 59.0 | 0.53 | 67.5 | 0.63 | 57.2 | 0.52 | 52.9 | 0.46 | 72.1 | 0.68 |
| S + To | 56.3 | 0.51 | 62.8 | 0.57 | 71.5 | 0.68 | 59.6 | 0.55 | 59.7 | 0.54 | 74.2 | 0.71 |
| S + To + Tx | - | - | 63.6 | 0.58 | - | - | 59.9 | 0.55 | 63.8 | 0.59 | 74.7 | 0.71 |
| S + To + Tx + G | 58.7 | 0.54 | 64.5 | 0.59 | 72.0 | 0.68 | 60.8 | 0.56 | 64.8 | 0.60 | 74.7 | 0.71 |
| All | 13.7 | 0.11 | 64.4 | 0.59 | 67.8 | 0.64 | 24.1 | 0.20 | 65.4 | 0.61 | 70.7 | 0.67 |

[1] Method of classification: ML: Maximum Likelihood; SVM: Support Vector Machine; DT: Decision Tree. [2] Accuracy values: OA: overall accuracy (%), K: Kappa coefficient. [3] Best combination of each group of thematic variables: S: spectral; S + To: spectral + topographical; S + To + Tx: spectral + topographical + textural; S + To + Tx + G: spectral + topographical + textural + geometrical; All: all variables.

The DT classifications yielded more accurate results than the ML or SVM approaches in all analyses performed. Independently of the different seasons studied or the most suitable combination of variables of each group, the DT classifier offered a 10–13% range of higher OA values than the other classification algorithms. Specifically, in the winter-13 image and considering the most efficient S + To + Tx combination, the OA difference between ML and DT exceeded 20% due to the poor performance of the textural variables in the ML classification. The use of textural information with different classifiers resulted in pronounced variations of accuracies (data not shown). The *OA* observed when all textural variables were included in the ML classification process was 35.9%. Accuracy improved when some textural variables were introduced separately, showing *OA* values of 49.1% for entropy and 50.8% for both correlation and mean. Nevertheless, the inclusion of homogeneity in the analysis resulted in an OA value of 11.8%. This behavior was the opposite of the obtained with the DT classifier, which reached the highest accuracy, 71.1%, combining only the homogeneity of all textural variables.

Table 6 summarizes the PA for every individual landform for the different images and classification methods considered. PA varied considerably according to the class classified. A general examination of the individual landform classification shows that 10 of the 13 landforms reached PA values higher than 75% in at least in one classification. Nevertheless, most classes showed high differences in PA values among classification methods and between images. Considering the average of all classifications, the 13S landform was the most accurately distinguished having a minimum PA of 79.5% with the ML algorithm in the summer-14 image, a maximum PA of 90.1% with the SVM classifier in the winter-13 image and mean value of 84.9%. Slightly higher differences in PA values were found in the 12S and 5P landforms, with mean values of 70% and 78.3% and a difference between maximum and minimum PA of 20.9% and 29.0%, respectively. The other landforms presented very prominent variations between

classifiers and/or images. For example, discrimination of the 2P landform was very successful when applying the ML method in the summer-14 image with a PA of 95.5%, but was also the landform that obtained the less accurate PA overall, with a value of only 0.2% with the SVM algorithm in the winter-13 image.

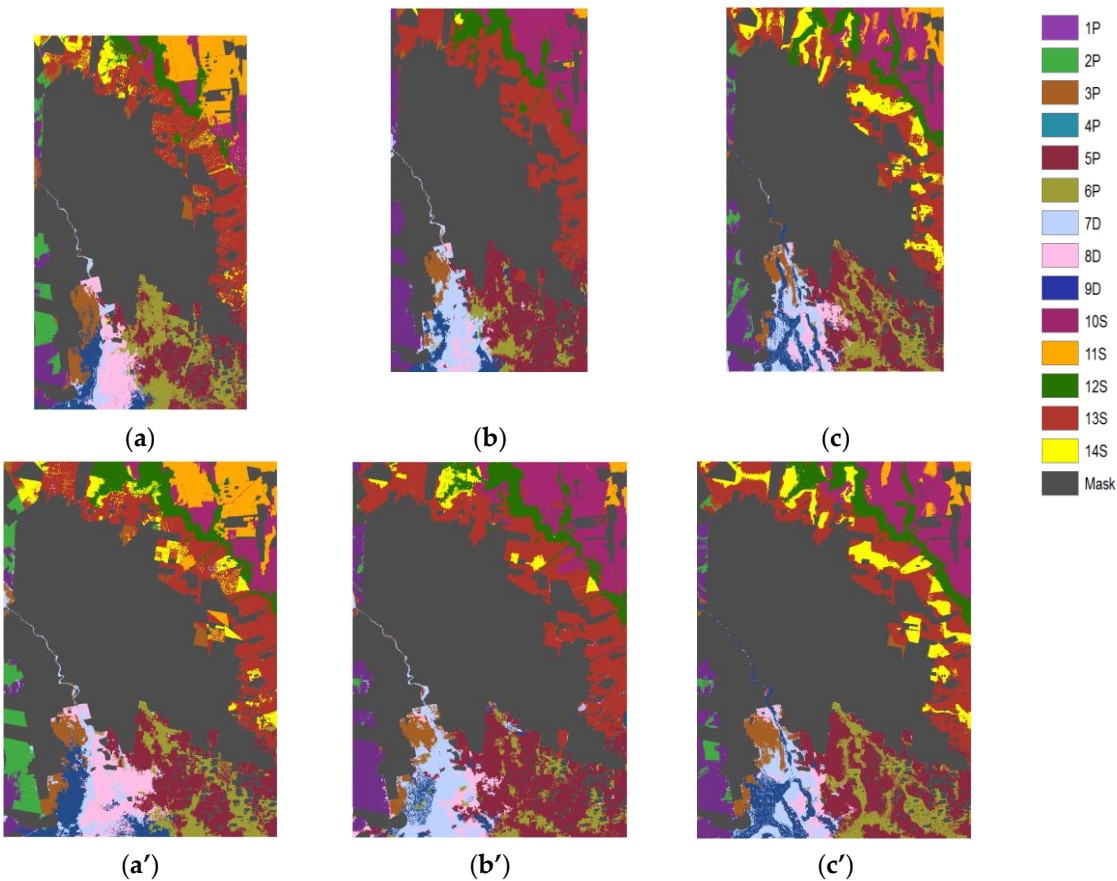

**Figure 5.** Result of the most accurate landform classifications obtained with the winter-13 image (**a–d**) and the summer-14 image (**a′–d′**): (**a**) Maximum Likelihood (ML), (**b**) Support Vector Machine (SVM), (**c**) Decision Tree (DT).

**Table 6.** Producer's accuracy (%) of the three classifications methods obtained with the best combination of object-based and topographical variables in winter and summer images using Maximum Likelihood, Support Vector Machine and Decision Tree algorithms.

| Method [1] | Imagery | Individual Landform Uses | | | | | | | | | | | | | Statistics [2] | |
|---|---|---|---|---|---|---|---|---|---|---|---|---|---|---|---|---|
| | | 1P | 2P | 3P | 5P | 6P | 7D | 8D | 9D | 10S | 11S | 12S | 13S | 14S | $\overline{x}$ | σ |
| ML | Winter-13 | 59.8 | 86.1 | 75.0 | 59.3 | 66.8 | 69.9 | 73.3 | 22.6 | 31.9 | 97.9 | 57.5 | 82.2 | 23.6 | 62.0 | 23.5 |
| | Summer-14 | 32.1 | 95.5 | 65.7 | 71.1 | 54.1 | 82.2 | 77.1 | 31.2 | 39.8 | 92.7 | 74.1 | 79.5 | 32.6 | 63.7 | 23.2 |
| SVM | Winter-13 | 96.0 | 0.2 | 45.6 | 83.6 | 27.2 | 52.2 | 30.2 | 68.1 | 90.7 | 14.8 | 64.7 | 90.1 | 2.4 | 51.2 | 34.1 |
| | Summer-14 | 89.5 | 10.5 | 53.4 | 83.4 | 31.1 | 33.7 | 30.5 | 68.3 | 88.1 | 29.9 | 68.4 | 84.8 | 17.5 | 53.0 | 28.8 |
| DT | Winter-13 | 75.5 | 91.2 | 72.5 | 88.3 | 33.8 | 62.7 | 66.0 | 64.6 | 78.8 | 61.1 | 78.4 | 84.1 | 48.4 | 69.6 | 16.1 |
| | Summer-14 | 90.4 | 54.6 | 70.3 | 84.3 | 51.4 | 63.6 | 62.1 | 64.1 | 92.4 | 29.7 | 77.0 | 88.7 | 49.0 | 67.5 | 18.8 |
| Statistics | $\overline{x}$ | 73.9 | 56.4 | 63.8 | 78.3 | 44.1 | 60.7 | 56.5 | 53.2 | 70.3 | 54.4 | 70.0 | 84.9 | 28.9 | | |
| | σ | 24.3 | 42.2 | 11.7 | 11.0 | 15.7 | 16.5 | 21.0 | 20.6 | 27.2 | 35.2 | 8.0 | 4.0 | 18.2 | | |

[1] Method of classification: ML: Maximum Likelihood; SVM: Support Vector Machine; DT: Decision Tree. [2] Basic statistical parameters: $\overline{x}$: mean; σ: standard deviation.

When comparing the performance of the three classification algorithms, the average *PA* values of all landforms considered jointly showed that DT method reached the highest values with 69.6% and 67.5% for the winter-13 and summer-14 images, respectively. Close *PA* values were observed

with the ML classifier, having a PA of 63.7% and 62.0% for the summer-14 and winter-13 images, respectively. Although the SVM classifications obtained better overall accuracies than the ML classifications, SVM showed the lowest average PA results, with 53.0% and 51.2% *PA* for summer-14 and winter-13, respectively.

### 4.3. Evaluation of the Importance of the Variables in the Prediction

Results obtained from the first analysis to determine which variables were to be used to obtain the most accurate classifications and their weight in each classification are shown in Table 7. From the 34 variables used to predict landform units, it was observed that only half of the pre-selected metrics were determined to be useful in at least one of the more accurate classifications tested, showing slight differences among their use depending on the different season images and the classifiers. The spectral variables *Mean* and Standard *Deviation* and the topographical variables *Elevation* and Altitude about Channel Network showed 100% average utility due to being selected in all the more accurate combinations, while and Channel Network Base Level offered 83.4%. Contrary to these variables, 12 of the 34 variables, particularly topographical and geometrical variables, were not used in any the classifications analyzed.

The spectral variable group showed more use of its own variables in the classifications, with the higher utility values ranging from 40% to 100%. Due to the fact that some of their variables were never chosen for the more accurate combinations, the topographical and geometrical variable groups presented lower utility values. The utility of the topographical variable group resulting in values from 27.3% to 54.5%, while the geometrical variable group exhibits utility values from 7.1% to 42.9%. The textural variable group exhibited the most erratic behavior, with a zero percent utility in two of the classifications and a utility of 100% in the remaining analysis.

From the point of view of the classifier, the number of variables used varied considerably. While the DT algorithm needed the lowest number of variables, 8, to reach its highest accuracy, the SVM classifier needed approximately double the variables, 15 and 17 for 2013 and 2014 imagery, respectively.

The second analysis was focused on analyzing the importance of each variable in the classification using only the DT information, where each variable intervenes with a different utility in the prediction model. The percentage of use for each variable can be observed in Figure 6. Clearly, in the DT classifications, three topographic variables were the most useful variables, with a high rate of appearance in the nodes of the trees. *Channel Network Base Level* and *Altitude about Channel Network* showed the highest percentages, with values of 25.7% and 18.8% for the winter-13 image, and slightly lower values of 19.7% and 12.7%, for the summer-14 image, respectively. *Elevation*, the other topographic variable used, presented more homogeneous values of about 12% in both images. Other variables observed in both trees but with lower importance in the process were the spectral variables *Mean* and *Standard Deviation*. Both of these variables, as with *Skewness* and all the textural variables, are calculated for every band, resulting in 7 bands for each variable. As seen in Figure 6, the percentage of both variables in all bands were considerably lower than the three topographical variables, with a maximum of 2.7% in bands 5 and 7 for the *Standard Deviations* variable for the winter-13 image and a maximum of 4.3% in band 5 for the *Mean* variable for the summer-14 image. Likewise, for *Skewness* and *Entropy* for the summer-14 tree, the highest *Mean* and *Standard Deviation* values were observed mainly in band numbers 5, 6 and 7, corresponding to near infrared and shortwave infrared bands. Other distinguished variables were *Rectangular Fit* and *Asymmetry*, both geometrical variables. *Rectangular Fit* was used in the 16.1% of the prediction model nodes for the winter-13 image, whereas *Asymmetry* was observed in 11.3% of the summer-14 tree nodes.

The results obtained in both analyses indicate the complexity of the predictions when considering the variation of the resulting spectral information from the satellite imagery collected in the different seasons and the behavior of the classification algorithms tested.

**Table 7.** Importance of the variables depending on their degree of use with the most accurate combination of object-based and topographical variables in winter and summer images using Maximum Likelihood, Support Vector Machine and Decision Tree algorithms.

| Type | Variable | Winter-13 Image | | | | Summer-14 Image | | | | Average |
|---|---|---|---|---|---|---|---|---|---|---|
| | | ML | SVM | DT | Utility (%) | ML | SVM | DT | Utility (%) | Utility (%) |
| Spectral | Mean | *1 | * | * | 100 | * | * | * | 100 | 100 |
| | St_Dev | * | * | * | 100 | * | * | * | 100 | 100 |
| | Skew | | * | | 33.3 | * | * | * | 100 | 66.7 |
| | Bright | | | | 0 | * | | | 33.3 | 16.7 |
| | Max_Diff | * | * | | 66.7 | * | * | | 66.7 | 66.7 |
| | Utility (%) | 60.0 | 80.0 | 40.0 | | 100 | 80.0 | 60.0 | | |
| Topographical | Elev | * | * | * | 100 | * | * | * | 100 | 100 |
| | Slope | * | | | 33.3 | * | | | 33.3 | 33.3 |
| | Aspect | * | | | 33.3 | | * | | 33.3 | 33.3 |
| | Plan_Cur | | | | 0 | | | | 0 | 0 |
| | Prof_Cur | | | | 0 | | | | 0 | 0 |
| | Alt_Ch | * | * | * | 100 | * | * | * | 100 | 100 |
| | Catch_Area | | | | 0 | | | | 0 | 0 |
| | Ch_Net | * | * | * | 100 | | * | * | 66.7 | 83.4 |
| | Conv_I | * | | | 33.3 | * | * | | 66.7 | 50.0 |
| | LS_Factor | | | | 0 | | | | 0 | 0 |
| | Wet_I | | | | 0 | | | | 0 | 0 |
| | Utility (%) | 54.5 | 27.3 | 27.3 | | 36.4 | 45.5 | 27.3 | | |
| Textural | GLCM_C | | | | 0 | | * | | 33.3 | 16,7 |
| | GLCM_E | | | | 0 | | * | * | 66.6 | 33.3 |
| | GLCM_H | | * | | 33.3 | * | * | | 66.6 | 50.0 |
| | GLCM_M | | * | | 33.3 | | * | | 33.3 | 33.3 |
| | Utility (%) | 0.0 | 50.0 | 0.0 | | 25.0 | 100 | 25.0 | | |
| Geometrical | Area | | | | 0 | | | | 0 | 0 |
| | Length | * | * | | 66.7 | * | * | | 66.7 | 66.7 |
| | Width | | | | 0 | * | | | 33.3 | 16.7 |
| | Asymm | * | * | | 66.7 | | * | * | 66.7 | 66.7 |
| | Border_I | | * | | 33.3 | | | | 0 | 16.7 |
| | Compact | | | | 0 | | | | 0 | 0 |
| | Density | | | | 0 | | | | 0 | 0 |
| | Ellip_Fit | | | | 0 | | | | 0 | 0 |
| | Main_Dir | | | | 0 | | | | 0 | 0 |
| | R_Largest | * | * | * | 100 | | * | | 33.3 | 66.7 |
| | R_Smallest | | | | 0 | | | | 0 | 0 |
| | Rect_Fit | * | * | * | 100 | | * | | 33.3 | 66.7 |
| | Round | | | | 0 | | | | 0 | 0 |
| | Shape_I | | * | * | 66.7 | | | | 0 | 33.3 |
| | Utility (%) | 36.4 | 42.9 | 21.4 | | 14.3 | 28.6 | 7.1 | | |
| No. of variables used | | 13 | 15 | 8 | | 12 | 17 | 8 | | |

Asterix (*) indicates the variable was included in the most accurate classification process.

## 5. Discussion

### 5.1. Stage 1: Basic-Spectral Variable Classifications

An overview of the five classification methods applied to basic-spectral variables derived from the Landsat 8 imagery for landform classification showed that object-based analysis (OBIA) clearly outperformed pixel-based analysis, with increases of overall accuracy reaching up to 37% with the DT classifier. The results of the classification algorithms varied considerably, especially in OBIA analyses, where the DT algorithm yielded the most accurate results, followed by SVM and ML. Although Ballantine et al. [7] and Iwahashi and Pike [29] concluded that spectral data improves classification because of the increased distinction between topographically similar landforms, our study suggests that only using the spectral reflectance of the multispectral Landsat 8 bands did not

offer enough information to obtain adequate overall accuracies for landform classifications, even when using advanced techniques like OBIA and complex data algorithms. The use of hyperspectral data in this type of study could increase the accuracy of landform classifications, since hyperspectral data can be sensitive to spectral differences due to the spatial gradient of moisture and mineralogical size and composition of each landform [30].

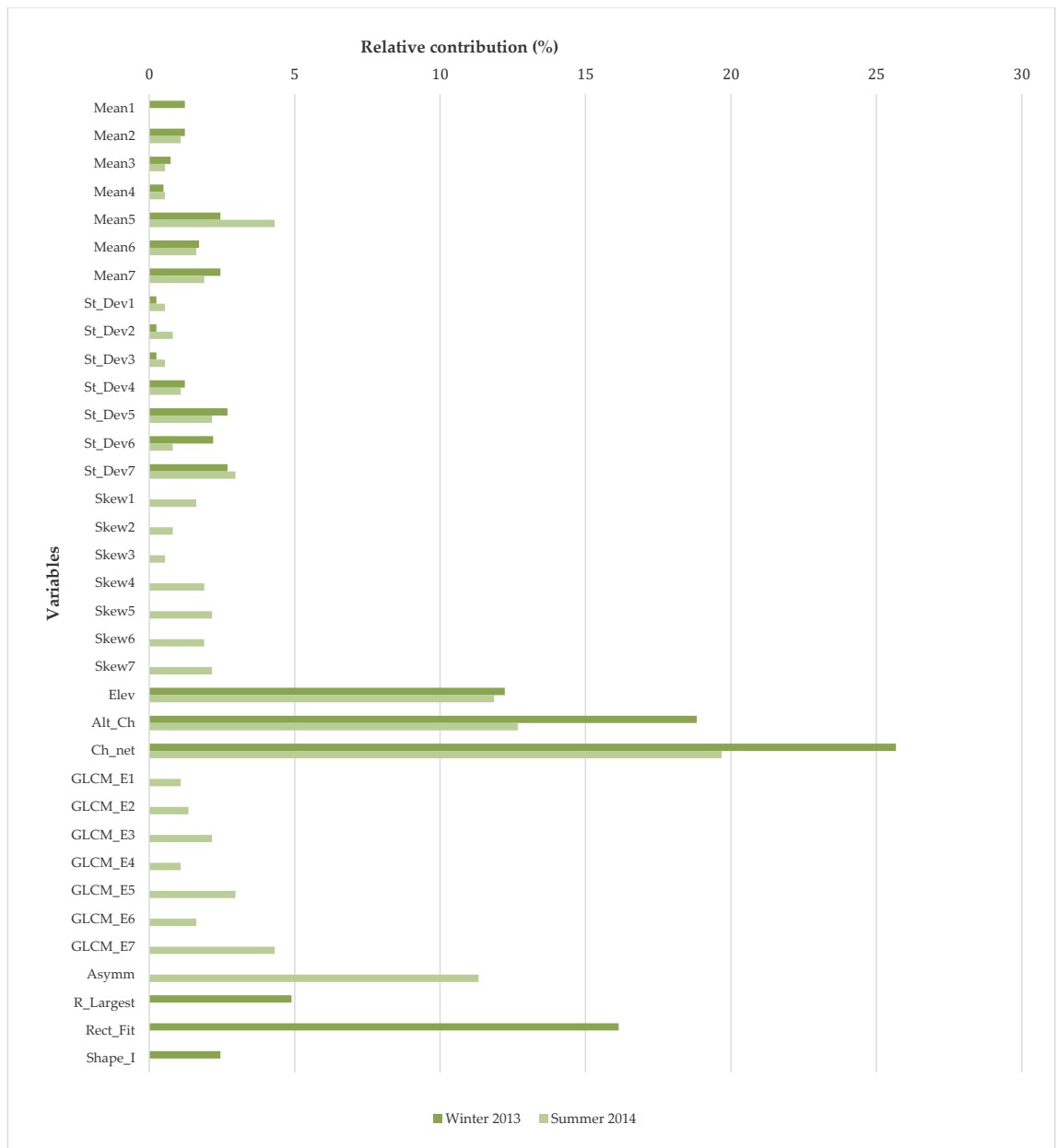

**Figure 6.** Relative contribution of advanced object-based and topographic variables for DT classifications performed with winter 2013 and summer 2014 Landsat 8 images. Variables described in Tables 2 and 3.

*5.2. Stage 2: Advanced Object-Derived + Topographic Variable Classifications*

In order to improve the classification results obtained previously, combinations of 5 spectral, 4 textural and 14 geometrical object-based derived variables together with 11 topographical derived variables were tested. From the results, it is clear that DT, a data mining technique, outperformed the other classifiers. Although DT yielded the highest classification accuracies out of all the classifiers

evaluated, it was also the most efficient, needing only 8 variables to reach the highest accuracy for both images. This optimization of the use of variables can be explained through its construction of knowledge modelling. DT approaches employ hierarchical recursive portioning of the data, resulting in decision rules that relate values or thresholds to the predictor variables with pixel classes [31,32]. When a large volume of predicted variables is introduced in a model, DT tends to be very efficient and robust, generally performing fast and being insensitive to noise in input data [31]. Because of this, it is one of the most commonly used algorithms in the machine learning and data mining communities and has become a de facto community standard against which every new algorithm is evaluated [33].

The other data mining classifier that performed well, although with less accuracy than DT, as the number of variables was very high (34 variables), was SVM. The fact that SVM yielded similar or even higher accuracies when all variables were introduced to the model with an optimum combination of variables shows that this algorithm was very robust in high dimensionality [34,35]. In this study, SVM needed more variables, 15 for winter-13 and 17 for summer-14, to generate the most precise combination. Similar accuracies were observed when all variables (34) where introduced in the model, compared to a reduction of variables. The optimal selection in this study showed that the larger the number of input variables for selecting samples are, the more the classifier is able to develop accurate hyperplanes [36], and that reducing the number of variables can have a negative effect on the sample set, which can lead to diminished classification results [37,38]. DT and SVM, both nonparametric classifiers, outperformed the ML classification because their ability to cope with non-normal distributions of input data and to accept a wide variety of input data in the form of both continuous and/or categorical variables [39].

ML was recognized as a stable and robust classification method and is currently one of the most widely used methods in classifying remotely sensed data [40]. For the ML algorithm, the use of prior probabilities for training data is considered important in environments where some classes are dominant spatially [41], but this is a disadvantage when having limited information beforehand.

Some of the predicting variables were very useful in distinguishing landforms from each other in the study area. Although each classification algorithm performed better with different combination of variables, clearly the spectral and topographical group of variables were the most valuable in most of the analyses. The textural and the geometrical group of variables were secondary, providing only minor further enhancements, but important in reducing uncertainties in classifications. Topographical data were important to derive morphographic and morphometric attributes, which are used in soil-landscape characterization at regional scales [9], and had been observed to be useful in hilly terrain. The elevation data, with an average utility of 100% in all classifications and a relative contribution around 12% in both DT classifications, could offer valuable information in the two faults observed in the study area, a main fault in the west and a secondary fault in the east. To that last point, however, in flat areas the *Elevation* variable alone do not facilitate interpretation of soil variations [42]. Hydrological variables such as *Altitude about Channel Network* and *Channel Network Base Level* were very useful in a territory that shows great influence of two rivers, where water bodies, main rivers, streams and meanders define the landscape. The different relative contribution of that hydrological variables in both images can be explained because the season in which the images are collected and, therefore, how the vegetation phenology could influence the other groups of variables directly or indirectly [43]. For the winter-13 image, collected during the dry season, characterized by the scarcity of vegetation and a greater visibility of surface terrain, the need for spectral variables was slightly lower than summer-13 and the use of textural variables were practically non-existent. With the DT analyses, where each variable intervenes with a different utility in the prediction model, the effect of the season is clearer. The high importance of topographic variables in the winter-13 DT classification, having more than 56% of the weight of the total variables, may be due to the increased topographical differences when the vegetation and water levels are reduced as well as the contour of the surface highlights. The spectral information was important in this study, particularly the infrared bands. The DT analyses showed the near infrared and the short-wave infrared bands were the most useful. The near infrared band is

very sensitive to vegetation changes, which are closely related to the type of landform. Regarding the bands of the shortwave infrared, these bands are related to water content characterization [30]. Both, native vegetation and moisture are very valuable information in a semi-arid ecosystem as the low elevation areas are more susceptible to moisture stress than the high elevation areas due to higher temperatures and less precipitation at low elevations [44]. Finally, although the geometrical variables were secondary in this study, the DT classifications showed a significant contribution of some of them, especially the rectangular fit variable, very useful for classification of man-made features. That can be explain due to the undergoing transformations from traditional land use to commercial agriculture, as this transformation is planned before the change. The area design to be altered but with native vegetation was included in the analysis and presented man-made divisions that forced to create regular objects in the segmentation process that not present the same contour (shape and size) than natural landforms.

In OBIA, the optimization of segmentation parameters may improve the accuracy of the final map. As these parameters, especially the scale parameter, define the size of the objects, Anders et al. [45] argued that segmentation and classification of all feature types at once may not give satisfying results when used for mapping entire areas of particular (complex) landscapes. For that reason, Anders et al. [45] performed stratified approaches depending on the characteristics of the landscapes, which yielded an average accuracy of 71%. This study, without the stratified OBIA classification but with more complexity in the use of variables (advanced object-derived and topographic variables) and with a data mining classification algorithm (DT), obtained similar accuracies, with OA of 72% and 74.7% for classifications of the winter-13 and summer-14 images, respectively.

## 6. Conclusions

The analysis performed provides new insights into the way combinations of advanced object-derived and topographic data with complex classification algorithms could be useful for mapping large and complex geomorphic areas with a variety of landscapes. The development of an accurate landform distribution map is feasible when an optimal combination of variables is classified with DT algorithms. The results presented here show that the choice of the type of classification, pixel- or object-based classification, and the choice of the classification algorithm are very important in order to obtain higher accuracies, especially when a large volume of variables is included in the analysis. In this study, object-based image analysis (OBIA) clearly outperformed pixel-based analysis, with increases of overall accuracy of up to 37% with the DT classifier. The data mining algorithms tested, SVM and DT, outperformed the other classifiers in all scenarios analyzed. For example, while in advanced object-based classifications increases of overall accuracy around 13.5% could be observed between DT and ML algorithms, in basic spectral object-based classifications, accuracies increased up to 49% between the SAM and DT algorithms. Regarding variables, not all offered valuable information in the classifications. Although some variables did not perform well with all classifiers, clearly, some topographical variables together with some spectral variables derived from the segmentation of the original Landsat imagery offered the most valuable information to the classifications. The topographical variables *Elevation*, *Altitude about Channel Network* and *Channel Network Base Level* and the spectral object-based derived variables *Mean* and *Standard Deviation* can be considered essential in this type of study. The implementation of these techniques together with the knowledge of contextual information of landscapes and soil forming factors could contribute to soil mapping at appropriate scales in areas of agricultural expansion for land evaluation and planning. As Taramelli [46] suggested, the morphometric analysis does not accurately map the landforms, but can be used as a first highlight of them. Therefore, to carry out an OBIA classification with the DT algorithm and including the five variables previously mentioned can automate the landform classification process in order to delineate the preliminary landform units. Although more accurate maps can be obtained combining other physical or chemical variables from the soil profiles, the reduction of work needed to develop this type of maps is considerable, especially in large areas.



**Author Contributions:** I.L.C.-G, C.A. and M.S.d.l.O. conceived and designed the experiments; I.L.C.-G, C.A., A.G.-F. and M.S.d.l.O. performed the experiments; I.L.C.-G. and C.A. contributed with analysis tools; I.L.C.-G. and C.A. analyzed the data; I.L.C.-G. writing—original draft preparation; I.L.C.-G, C.A., A.G.-F and M.S.d.l.O. writing—review and editing.

**Funding:** This research received no external funding.

**Acknowledgments:** We express our gratitude to the National Institute for Agricultural Technology (INTA), Argentina, for sharing information.

**Conflicts of Interest:** The authors declare no conflict of interest.

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
