# Peer review of "Combining Object-Based Image Analysis with Topographic Data for Landform Mapping: A Case Study in the Semi-Arid Chaco Ecosystem, Argentina"

_ijgi, doi:10.3390/ijgi8030132_

Round 1
Reviewer 1 Report
The paper is well written and methodology is well explained. The approach of segmentation and optimization of object variable are also explained well. The hypothesis i.e. to map landforms which control development of soil types is good. I have few comments only.
Line 28 – “…….landforms and even, of types of soil.” It should be “…….landforms and even types of soil.”
Line 39 - “……..informed planning of land use. That situation…”It should be “……..informed planning of land use, which is……”
Line 84 – Is there any specific reason of using old L8 data. Why recent L8 data were not used?
Table 1 –“ Distal megafal” should be “Distal Megafan”
Authors need better review of work on application of OBIA for landform mapping. Following are some recent good studies on this topic.
1. Tapas R. Martha, A. Mohan Vamsee, Vikas Tripathi, K. Vinod Kumar - Detection of coastal landforms in a deltaic area using a multi-scale object-based classification method, CURRENT SCIENCE, VOL. 114, NO. 6, 25 MARCH 2018
2. Gasiganti T. Vamshi, Tapas R. Martha, K. Vinod Kumar-An object-based classification method for automatic detection of lunar impact craters from topographic data, Advances in Space Research 57 (2016) 1978–1988
3. Dragut, L. and Eisank, C., Automated object-based classification of topography from SRTM data. Geomorphology, 2012, 142/141, 21–33.
Section 3.1.1 – It is not clear what are the bands of L8 used in Multiresolution segmentation.
Table 5- It got altered. Difficult to read.
Authors have concluded that OBIA outperform PBIA. But one of the strengths of OBIA i.e. contextual information of objects was not considered in the classification. Can we think that, if contextual parameters (e.g. association of terraces, megafans etc) are considered, then accuracy will be high. Authors may comment on this.
Figure 3 – Authors may explain how rectangular fit has significant contribution for classification of natural features (e.g. landforms). Normally, rectangular fit is useful for classification of man-made features.
Authors need to add few figures showing comparison of classified landform and reference landform. If maps showing improvement in classification with various methods can be added, then it will be good for readers.
Author Response
We thank the reviewers for their valuable comments, which have enabled us to significantly improve this paper. We carefully revised the manuscript. Please find our responses to each individual comment in the attached file.

Reviewer 2 Report
It is important to map the soil and terrain of the region for the sustainable development. And the increasing availability of high-quality digital data is gradually replacing classical techniques of soil and terrain mapping. In this study, advanced object-based variables derived from Landsat 8 images and topographic variables were used to map the landform of Chaco ecosystem in Argentina with different classification algorithms. The results indicate that Decision Tree is the most accurate classifier.
Although the study itself is of valuable, a major revision would be necessary before acceptance for publication in this journal.
Comments:
1. Line 110-115. Training samples selection is very important for classification. The samples should be representative and have balanced spatial distribution. Please give out the sample distribution map of different landform.
2. Line 203. Please gave out the equation of OA, K and PA.
3. The analysis of classification results takes a large part to describe the data, but lacks the analysis of variables impact and algorithm mechanism. For example, DT algorithm yielded the highest classification accuracies needing only 8 variables for both images, while the other algorithms need more variables. Do these variables overlap in different classification algorithm? And are these overlapped variables related to the landform? The reduction of the number of the variables need to be related to the mechanism of the algorithm or the influence of the variables or anything else? And what is the basis of support? Are 8 variables needed in this study to yield the highest accuracy universal or limited to the landform of the region? If they are limited to the landform, what kind of relationship between the variables and the landform of the region? Please answer the above questions in details. I think this is the core part of the study, not just the description of the data.
4. There are many spelling mistakes in this article, such as
Line 72 “8800 km2”;
Line 384 “……a de facto……” ;and etc.
Please correct them.
Author Response

(The authors gave the same response as above.)

Reviewer 3 Report
Reviewer (anonymous):
This is an article not suitable for publication in International Journal of Geo-Information" as it is. Main reasons:
* the scope is very narrow - detecting topographic data for landform mapping from image and SRTM data is not a new issue; the authors present it as a complex landform including its boundaries (Taramelli A., (2011), Detecting Landforms Using Quantitative Radar Roughness Characterization and Spectral Mixing Analysis, In B. Murgante et al. (Eds.): Geocomputation, Sustainability & Environ. Planning, SCI 348, pp. 225–249.; Zhu, A.X.: A similarity model for representing soil spatial information. Geoderma 77, 217–242 (1997)
Zhu, A.X.: Measuring uncertainty in class assignment for natural resource maps using a similarity model. Photog. Eng. and Rem. Sens. 63, 1195–1202 (1997)), but in geomorphology landform boundaries are far the most 'simple' to be recognized, also using traditional methods and new tools. This cannot be found back in the article. This is a serious flawn.
* no concise goal/objective is formulated and no real research questions are mentioned in the introduction, it remains too vague. If we talk about landform we need a clear definition in the abstract and a clear introduction to this end.
* important lack of structure and content, examples:
- abstract is a reflection of content showing no discussion linked to the landform that Is detecting.
- under methods I expect a diagram showing a floch chart of the different approach still related to what in litterature is really present;
* geomorphological terminology is used in a poor sense when used; many important geomorphological maps are known, I do not see a refelction of that, either in the text + figures, as in the reference list conisdering the area under reserach.
* There is real link to the scientific importance and the societal importance. For what is detection/insight in landforms like this necessary? The 3D (in depth) dimensions are perhaps even more important (geophysical techniques, material differences etc., intercalatios with the floodplain deposits and more...)
* I do have another major concern - I ran a check on GEOREF from the present to 2002 and came up with a huge number of references on landform detection using earth observation methodology. I want to be assured that your lack of citations from this period means none of thosse contain material that would support what you are doing. Are you aware of the huge body of work that were in the last 10 years implemnetd about spectral signal?
Just for an example and references within them:
- Manzo C., Valentini E., Taramelli A, Filipponi F., Disperati L. (2015), Spectral characterization of coastal sediments using Field Spectral Libraries, Airborne Hyperspectral Images and Topographic LiDAR Data (FHyL), International Journal of Applied Earth Observation and Geoinformation 36, 54–68
- Valentini, E.; Taramelli, A.; Filipponi, F.; Giulio, S.; (2015), An effective procedure for EUNIS and Natura 2000 habitat mapping in estuarine ecosystems integrating ecological knowledge and remote sensing analysis, Ocean and Coastal Management, 108: 52-64, http://dx.doi.org/10.1016/ j.ocecoaman.2014.07.015
* Classiffication results. I expect concise listing of results including some figures/tables? That disdcuss with a landform related geomorphological terminology. If this is not going to be introduced I do see the main aim of the research exept pure technicality, it is more like a techical discussion were the objective looks like a typical a methodological issue.
So I invite the authors to make a choise: what is the final scope of the paper? Detecting landform deserve a landform discussion not only a comparison of different approach that is not vaidated in terms of geomorphology.
Author Response

(The authors gave the same response as above.)

Round 2
Reviewer 2 Report
The authors have made great efforts to improve the quality of this manuscript. And I think the submission has been greatly improved and is worthy of publication.
Author Response
Once again we thank the reviewers for their valuable comments, which have enabled us to significantly improve this paper during the revision process.
Reviewer 2
Comments and Suggestions for Authors
The authors have made great efforts to improve the quality of this manuscript. And I think the submission has been greatly improved and is worthy of publication.
Thank you for your suggestions and comments. They were very valuables.

Reviewer 3 Report
Dear Authors the new submission is now suitable for the publication. I do still suggest the authors to try to better define the landform with more detailed eplanation. It will be a value added point for the discusion.
Author Response
Once again we thank the reviewers for their valuable comments, which have enabled us to significantly improve this paper during the revision process.
Reviewer 3
Comments and Suggestions for Authors
Dear Authors the new submission is now suitable for the publication. I do still suggest the authors to try to better define the landform with more detailed eplanation. It will be a value added point for the discusion.
Thank you for the recommendation. The suggestion has been taken into account and more information was included in the new version.
Thank you for your suggestions and comments. They were very valuables.
